# Mammalian Animal and Human Retinal Organ Culture as Pre-Clinical Model to Evaluate Oxidative Stress and Antioxidant Intraocular Therapeutics

**DOI:** 10.3390/antiox12061211

**Published:** 2023-06-03

**Authors:** Martina Kropp, Mohit Mohit, Cristina Ioana Leroy-Ciocanea, Laura Schwerm, Nina Harmening, Thais Bascuas, Eline De Clerck, Andreas J. Kreis, Bojan Pajic, Sandra Johnen, Gabriele Thumann

**Affiliations:** 1Experimental Ophthalmology, University of Geneva,1205 Geneva, Switzerlandnina.harmening@unige.ch (N.H.); bojan.pajic@orasis.ch (B.P.); 2Department of Ophthalmology, University Hospitals of Geneva, 1205 Geneva, Switzerland; 3Hôpital Privé La Louvière, 59042 Lille, France; 4Cabinet Ophtalmologie Sébastopol, 59000 Lille, France; 5Department of Ophthalmology, University Hospital Rheinisch-Westfälische Technische Hochschule (RWTH) Aachen, 52074 Aachen, Germanysjohnen@ukaachen.de (S.J.); 6Eye Clinic ORASIS, Swiss Eye Research Foundation, 5734 Reinach, Switzerland; 7Department of Physics, Faculty of Sciences, University of Novi Sad, 21000 Novi Sad, Serbia; 8Faculty of Medicine of the Military Medical Academy, University of Defense, 11000 Belgrade, Serbia

**Keywords:** retina organ culture, neuroretinal degenerative disease, age-related macular degeneration (AMD), diabetic retinopathy (DR), oxidative stress, antioxidant, scutellarin, PEDF, GM-CSF, 4R

## Abstract

Oxidative stress (OS) is involved in the pathogenesis of retinal neurodegenerative diseases such as age-related macular degeneration (AMD) and diabetic retinopathy (DR) and an important target of therapeutic treatments. New therapeutics are tested in vivo despite limits in terms of transferability and ethical concerns. Retina cultures using human tissue can deliver critical information and significantly reduce the number of animal experiments along with increased transferability. We cultured up to 32 retina samples derived from one eye, analyzed the model’s quality, induced OS, and tested the efficiency of antioxidative therapeutics. Bovine, porcine, rat, and human retinae were cultured in different experimental settings for 3–14 d. OS was induced by a high amount of glucose or hydrogen peroxide (H_2_O_2_) and treated with scutellarin, pigment epithelium-derived factor (PEDF), and/or granulocyte macrophage colony-stimulating factor (GM-CSF). The tissue morphology, cell viability, inflammation, and glutathione level were determined. The retina samples showed only moderate necrosis (23.83 ± 5.05 increased to 27.00 ± 1.66 AU PI-staining over 14 d) after 14 days in culture. OS was successfully induced (reduced ATP content of 288.3 ± 59.9 vs. 435.7 ± 166.8 nM ATP in the controls) and the antioxidants reduced OS-induced apoptosis (from 124.20 ± 51.09 to 60.80 ± 319.66 cells/image after the scutellarin treatment). Enhanced mammalian animal and human retina cultures enable reliable, highly transferable research on OS-triggered age-related diseases and pre-clinical testing during drug development.

## 1. Introduction

Aging is seen as a disruption of homeostasis and affects all organs [1]. One key factor is oxidative stress (OS), which is induced by an imbalance of reactive oxygen species (ROS) and a weakened antioxidant cellular defense, though, in normal aging, an excess of ROS can be still tackled [2]. Nevertheless, this imbalance, together with mitochondrial dysfunction, damaged protein accumulation, epigenetic alterations, telomere shortening, and abnormal intracellular signaling, result in the frailty syndrome, which is defined by a lowered physiological reserve and reduced resistance to stressors [3]. In accelerated aging, ROS production can no longer be controlled, and age-related diseases, such as cardiovascular diseases, metabolic disorders, cancer, and neurodegeneration, are the consequence [1,2,3]. Particularly in industrialized countries, health care has improved, and consequently life expectancy has increased [4,5]. Thus, the prevalence of age-related diseases is increasingly significant and has become a severe burden in terms of the quality of life of patients, health care, and socio-economic systems (as analyzed, e.g., by Bae et al. for Korea [6]).

Age-related macular degeneration (AMD) and diabetic retinopathy (DR) belong to this group of OS-triggered, age-related disorders [7,8,9]. AMD is the major cause of blindness in elderly patients in industrialized countries, and DR is the main reason of severe vision loss in the working age population [10]. In contrast to an imbalance of angiogenic and anti-angiogenic factors as seen in neovascular AMD (nAMD), the avascular form of AMD (aAMD) is mainly triggered by inflammatory processes and increased OS [11]. Bruch’s membrane and retinal pigment epithelial (RPE) cells become dysfunctional, and metabolic waste products (“drusen”) accumulate [11]. This breakdown of retinal homeostasis leads to RPE cell death, followed by photoreceptor death and finally retinal neuron cell death, cumulating in geographic atrophy (GA), the late stage of aAMD [11]. In DR, badly controlled hyperglycemia leads to microangiopathy, vessel occlusion, and leakage of the blood–retina barrier, causing hemorrhages, edema, and diminished retinal neuronal function; finally, it results in retinal neurodegeneration and cell death [12]. It has to be noted that the brain and the eyes are particularly prone for OS and the development of associated diseases due to their particularly high oxygen consumption, but, on the other hand, they also play a role in antioxidant defense [2,13]. A better understanding of the underlying pathomechanisms and treatment approaches for these diseases is of high demand.

Novel treatments have to be tested in vivo in animals to get approved by regulatory authorities; however, ethical issues and limits in transferability to patients make the development of alternative models necessary [14]. Cell cultures are limited to preliminary testing due to the lack of complexity and transferability of the systems [14,15]. Organ cultures can be performed using human tissue and perfectly combine the methodological simplicity of in vitro studies with the higher biological complexity of in vivo models [16]. Additionally, they save animals and are ethically preferable to in vivo projects (4R principles: refine, reduce, replace, responsible). For example, retina organ cultures allow us to analyze pathomechanisms including OS and test different treatments or doses of molecules on multiple samples from the same individual animal/human donor eye. It was first described in 1926 by Strangeways and Fells, who cultured 64–72-hour-old chick embryonic eyes up to 32 d using an extract from the embryos mixed in a balanced salt solution [17]. However, compared to a culture of fetal or newborn tissue, which contains potent, immature retinal progenitor cells, a culture of adult, mature retinae remains challenging despite the quality of the starting material, due to the special requirements of culture conditions. Murali et al. reported a cell culture of isolated neural retinal cells (from 31–89-year-old human donors) for up to 8 weeks as a suspension culture [18]. However, a culture of the complex retinal tissue structure (from adult donors) can be maintained only for 7 to 14 d; the lack of vascular perfusion, RPE, and connectivity to the brain via the optic nerve, as well as the continuous high demand of nutrients and oxygen, lead to degeneration by inflammation and OS [19]. Moreover, except for the findings of one report [20], retinal function, i.e., electric activity, can be recorded only up to 8 h post mortem [19,21,22]. Finally, cultures of aged and diseased human retinae—the tissue of interest in research on OS and ageing, useful for analyzing pathomechanisms and testing novel treatment approaches in disease models, are even more challenging. OS-induced damage and other imbalances of normal retinal homeostasis make the maintenance of tissue in cultures additionally difficult. Despite these challenges, important insights into retinal physiology [23,24] and pathology [25,26] have been made, and novel treatment options [27,28] been evaluated in retina organ cultures.

Our work focuses on the enhancement of adult/aged retina organ cultures as a preclinical test system for novel treatments against retinal neurodegeneration, such as the cell-based non-viral ex vivo gene therapy approach for treating AMD. To recover a cell-protective, antioxidant retinal environment, RPE or iris pigment epithelial (IPE) cells are transfected using the non-viral Sleeping Beauty (SB100X) transposon system with the genes coding for PEDF and GM-CSF [29,30]. PEDF is a cell-protective protein with multiple beneficial functions and has been evaluated, e.g., in models of osteosarcoma [31] and for photoreceptor and amacrine cell survival [32]. PEDF, ubiquitously expressed by RPE cells, is a molecule that has anti-angiogenic, anti-tumorigenic, and neurotrophic activities. It promotes the survival and functioning of neurons, protects them from oxidative damage, and inhibits apoptosis by the activation of the NF-κB signaling pathway [33]. GM-CSF is a multi-faceted factor expressed in RPE cells [34], responsible for the proliferation, differentiation, and maturation of myeloid cells and adaptive immune responses to inflammation and infection [35], especially in the central nervous system [36]. It counteracts oxidative damage by inhibiting apoptosis via the SRC-dependent STAT3 (signal transducer and activator of transcription 3) pathway, decreases BAD (Bcl-2 agonist of cell death) and increases BCL-2 (B-cell lymphoma 2) expression, and upregulates the production of neurotrophic factors [37]. Moreover, the phytochemical scutellarin, found in *Scutellaria Barbata*, *S. lateriflora*, and *S. baicalensis*, is tested as an antioxidant nutrient in DR [38]. This flavonoid is used in Chinese medicine with confirmed efficiency in, e.g., cardiovascular diseases and diabetic complications [38] due to its function as a potent radical scavenger as well as an inhibitor of induced nitric oxide synthase (iNOS) expression, lipid peroxidation, and cell death [38].

In the present work, different retina organ culture models are evaluated, and their advantages and drawbacks as preclinical test systems are discussed. OS-triggered degeneration is provoked by the high-glucose or H_2_O_2_ treatments, with our techniques continuously improving. The antioxidant and cell protective effects of scutellarin, PEDF, and GM-CSF are analyzed in retina cultures, confirming the suitability of these models as pre-clinical test systems in general and for OS-triggered retinal neurodegeneration in particular.

## 2. Materials and Methods

### 2.1. Animals and Cells

Globes from adult cattle and pigs from local slaughterhouses 1 to 6 h from the time of slaughter, transported on ice, were used for Models 1–3 and 5a. Adult rat retinae from 8-week-old Brown Norway rats (Charles River, L’Arbresle Cedex, France) were used for Model 4 immediately after their sacrifice. In the university’s animal facility (University of Geneva, Geneva, Switzerland), animals had access to food and water ad libitum. Euthanasia was performed under general anesthesia using ketamin (Ketalar^®^, Pfizer, Zurich, Switzerland; 100 mg/kg) and Xylazine (Rompun^®^, Bayer, Leverkusen, Germany; 10 mg/kg) diluted in NaCl (Merck, Darmstadt, Germany) at room temperature (RT), intra-peritoneally (i.p.) injected; then, Thiopental (Inresa, Freiburg, Germany; 150 mg/kg) diluted in NaCl was i.p. injected. Human donor eyes were purchased from the Lions Gift of Sight Eye Bank (Saint Paul, MN, USA) 2–8 d post mortem.

### 2.2. Tissue Isolation and Culture Conditions

#### 2.2.1. Bovine Retina–RPE Culture (Model 1)

The eyes were washed for 3 min with 70% ethanol (EtOH) (Bichsel, Interlaken, Switzerland) followed by ice-cold sterile phosphate buffer saline (PBS, Merck) containing 0.1% EtOH. The anterior segment was cut approximately 3.5 mm posterior to the limbus, and the posterior segment was flattened by a radial incision of the sclera using microscissors (Fine Science Tools, Heidelberg, Germany). With forceps (Fine Science Tools), the vitreous was carefully removed and the globe was sectioned into four pieces. Using a cell scraper (Milian, Boswil, Switzerland), the retina–RPE–choroid complex was separated from the underlying sclera and placed on a nitrocellulose membrane (pore size 0.45 µm, Schleicher & Schuell, Dassel, Germany) with the choroid facing downwards.

The retina–RPE–choroid complex was then sectioned into 5 × 5 mm samples using a surgical blade (Swann-Morton, Sheffield, UK). Nine samples were placed on customized, sterile, perforated, high-grade stainless-steel grid support benches in 12-well culture plates (Vitaris AG, Baar, Switzerland), and the wells were filled with serum-free DMEM (GIBCO, Eggenstein, Germany) supplemented with 2.8 mM L-glutamine (GIBCO), 673 U/mL penicillin, and 673 µg/mL streptomycin (Merck) (n = 36 from 4 retinas). All wells were filled with medium up to a level just below the upper retina–vitreous interface of the explant. Culture medium was changed every 24 h for a total culture duration of 4 d.

#### 2.2.2. Dynamic Porcine Retina Culture (Model 2)

Enucleated eyes were cleaned from rest of muscle tissue using forceps and microscissors (Fine Science Tools) and put into a beaker (Schott, Jena, Germany) filled with PBS (Merck). Then, globes were disinfected for 2 min in 70% EtOH (Bichsel) before they were transferred to a beaker (Schott) filled with sterile Ringer solution (B.Braun, Melsungen, Germany). For RPE–retina complex isolation, eyes were placed on a mull compress (Lohman & Rauscher, Rengsdorf, Germany) on a ceramic tile. Using loupe glasses (magnification 2.5×, Starmed, Grafing, Germany), the bulb was opened by a radial cut with a scalpel no. 11 (VWR, Dietikon, Switzerland) within the pars plana to ensure that retina and RPE stayed bound. Using forceps, the anterior part of the globe was lifted, the vitreous was grasped using Colibri forceps (Fine Science Tools), and both were separated from the posterior part and discarded. The nasal-orientated quarter of each retina (part of the visual streak and thus showing a high rod density) was always cut from the rest of the posterior part using forceps and microscissors. Drying out of the tissue was avoided by rinsing with Ringer solution. Under tension, the tissue was fixed on styrofoam with three canulae (Beckton & Dickinson, Franklin Lakes, NJ, USA). At the ora serrata, the RPE was detached from the capsule and held by Colibri forceps. By dissecting present vessels, a pocket was carefully prepared between capsule and RPE without lesioning RPE and retina. The white ring of the MinuSheet^®^ system (Minucells and Minutissue Vertriebs GmbH, Bad Abbach, Germany) [39,40] was pushed into the pocket, while the black ring was placed onto the tissue and the ring system gently closed using blunt forceps to avoid lesioning of the tissue. Excess tissue was cut, and the samples were placed in the MinuSheet^®^ chamber for perfusion culture. One chamber was filled with 6 samples, which were vertically oriented to the flow of medium, which was freshly prepared and stored for a max. of 3 d at 4 °C. During perfusion, the ice-cooled medium flew with a fixed flowrate of 56.5 µm/h through semi-permeable silicon tubes into the airtight closed Minuth chamber, passing a peristaltic pump (Fisher Scientific, Reinach, Switzerland). The flow rate was chosen to create a fluid pressure in the range of normal in vivo intraocular pressure (10–21 mmHg). The chamber and at least 10 cm of the tubes were placed on a 42 °C warmed heating plate to guarantee an inner-chamber temperature of 37 °C; for experiments with an inner-chamber temperature of 21 °C, perfusion took place at RT. Finally, the system was protected by a heat-conserving plastic cover. The media used left the perfusion chamber after passing the tissue and were collected in a glass bottle (Schott). The maximum culture duration was 3 d. The media used were DMEM (Biochrome, Cambridge, UK) or Ames medium (8.9 g/L, Merck), supplemented with 10 mL Amphotericin B (AMIMED, Allschwil, Switzerland) and 8 mL Penicillin/Streptomycin (Merck) for 1 L of medium. The Ames medium was additionally supplemented with 0.538 g NaCl (Merck), 0.634 g NaHCO_3_ (Merck), and 5.957 g HEPES (Merck).

#### 2.2.3. Static Porcine Retina Culture (Model 3)

The remaining muscle tissue and adnexa were removed with surgical scissors (Fine Science Tools). Then the globes were rinsed in ice-cold PBS (Merck), followed by disinfection in EtOH 70% for 2 min and rinsing in sterile ice-cold PBS. A scleral incision was made using a surgical blade around 3 mm posterior to the limbus and extended circumferentially with surgical scissors to dissect the anterior segment; the vitreous was kept in place to hydrate the retina and avoid RPE detachment. Next, the globe was cut into five equal pieces by cuts from the limbus to the optic nerve using scissors (Fine Science Tools). The samples were placed with the vitreous downwards onto a glass Petri dish; the retina was gently dissected from overlaying RPE, choroid, and sclera. Remaining RPE cells were washed off using BSS Plus (Alcon, Zug, Switzerland). A sterile, 15 × 15 mm small nitrocellulose membrane was placed on the photoreceptor layer, the tissue was turned, and the vitreous was removed. The tissue samples were put on customized, sterile, perforated, high-grade stainless-steel grid support benches (0.8 mm) in 12-well culture plates with the photoreceptors directed downwards; culture wells contained only 1.5 mL of medium, allowing the retinae to have contact with the air. Retinae were cultured for 1 to 4 d at 21 °C, with 5% CO_2_ and 95% humidity. Medium was not exchanged but was freshly added if the level decreased below the photoreceptor layer due to evaporation. The medium used was Ames (8.9 g/L, Merck) supplemented with 2 mM NaCl (Merck), 7.5 mM NaHCO_3_ (Merck), 25 mM HEPES (Merck), 1% Amphotericin B, and 0.8% Penicillin/Streptomycin.

#### 2.2.4. Rat Retina Culture (Model 4)

Due to the smaller size of the rat eye, tissue preparation was performed under semi-sterile conditions using a dissection microscope under a laboratory hood based on protocols from Bull et al., Kaempf et al., and Leroy-Ciocanea [37,41,42]. Immediately after sacrifice of the animals and enucleation, globes were washed in PBS followed by disinfection for 30 s in Betadine^®^ (Mundipharma, Frankfurt, Germany) before being rinsed again in sterile PBS. A scleral incision was made posterior to the limbus using a 25 G needle (Beckton & Dickinson, Franklin Lakes, NJ, USA) and extended circumferentially using Vannas scissors (Fine Science Tools) to dissect the anterior from the posterior segment. As explained in Model 3, the vitreous was kept to preserve the retina hydrated and to avoid detachment of the RPE layer. Using Vannas scissors, the retina was then cut into two halves by two cuts towards the optic nerve head. The samples were transferred to a glass Petri dish with the vitreous directed downwards, and using a small steel sooth spoon, RPE, choroid, and sclera were removed. A sterile, 10 × 10 mm small nitrocellulose membrane was placed on the photoreceptor layer of the retina. The complex was turned and placed on customized, sterile, perforated, high-grade stainless-steel grid support benches (0.8 mm) in 12-well culture plates with the photoreceptors directed downwards to allow air contact. Only 1.5 mL of medium was added to culture wells. The tissue was cultured at 21 °C with 5% CO_2_ and 95% humidity for 9–13 d.

Different medium conditions were tested to improve the culture system, but in any case, half of the medium was exchanged every other day. First, Ames medium as described above was used. Secondly, Neurobasal-A medium (ThermoFisher Scientific, Waltham, MA, USA) supplemented with the neurotrophic factors 2% B-27 and 1% N-2 (ThermoFisher Scientific) was tested. Third, Ames medium supplemented with 2% B-27 and 1% N-2 was added to culture wells. All media were supplemented with 10 mL Amphotericin B and 8 mL Penicillin/Streptomycin in 1 L of medium.

#### 2.2.5. Semi-Long Porcine and Human Retina Culture (Model 5a and 5b)

Porcine and human retinae were similarly treated to obtain cultures. As aforementioned, the globes were rinsed in PBS (10 mL), disinfected in Betadine^®^ (10 mL, 2 min) and washed in sterile PBS (10 mL, 2 min) for a second time after removal of remaining muscles and adnexa with surgical scissors. A flat scalpel no. 11 (VWR) was used to make a small incision around 2 mm below the limbus and extended with curved micro-scissors (Fine Science Tools). Cornea, iris, lens, and vitreous were removed using forceps before four cuts towards the optic nerve were made into the posterior part using scalpel no. 10 (VWR) to flatten the tissue. The retina was detached from the RPE by flushing the tissue with sterile PBS and was allowed to “slide” into a glass Petri dish (90 mm) filled with 10 mL sterile PBS with the photoreceptors directed downwards. A 6 mm small biopsy punch (Stiefel, Brantford, UK) was used to cut up to 32 samples per retina; the punch had to be positioned perpendicular to the retina and twisted a few times. The retina remained in the punch so it could be easily transferred to a culture insert (0.4 µm permeable, transparent PET membrane, Corning, Corning, NY, USA) prefilled with 1 mL sterile PBS. Then, the retina detached itself from the punch and settled down to the base by gravity when immersed in PBS; if necessary, the punch was moved up and down slowly to allow detachment. PBS was carefully removed with a pipette without touching the tissue before transferring the insert to a 12-well plate prefilled with 1 mL Ames medium and culturing it at 21 °C and 5% CO_2_ for 14 d; medium was exchanged every day. Explants were cultured in serum-free Ames medium (pH = 7.4) supplemented with growth factors 1% N2 and 1% B27 along with 1% antibiotics (Penicillin/Streptomycin) and 1% fungicide (Amphotericin B).

#### 2.2.6. Summary of Retina Models

The summary presented in Table 1 juxtaposes all retina culture models presented and compares key culture parameters. Figure 1 visualizes the different models showing the minuth chamber (Model 2), the customized stainless-steel grids (Models 1, 3, and 4), rat retina compared to porcine retina samples cultured, and key features of Model 5.

#### 2.2.7. Induction of OS by High-Glucose Conditions and Scutellarin Treatment

In Model 3, high-glucose conditions were induced to mimic diabetic conditions, and retinae treated with different concentrations of Scutellarin. Commercially available Ames medium contained 6 mM D-glucose (1.081 g), which was increased by addition of 24 mM D-glucose (Merck) to a final concentration of 30 mM to reach high-glucose levels. Scutellarin (Xie Qifan, Chengdu, China) was added in concentrations of 1, 10, or 100 µM. Other culture conditions were maintained.

#### 2.2.8. Induction of OS by H_2_O_2_ Incubation and PEDF/GM-CSF Treatment

OS was induced on retinae (Model 4) by adding 300 μM H_2_O_2_ (Merck) for 3 h on day 3. To test their neuroprotective and antioxidative effects, 500 ng PEDF (BioProductsMD, Middletown, MD, USA), 500 ng GM-CSF (Peprotech, London, UK), or both were added to the medium for 3 days; due to the short half-life of the proteins, the medium was exchanged every day with freshly added proteins from day 1 to day 3. Cultures were terminated on day 13.

### 2.3. Brightfield Microscopy of Flatmount Preparations and Degeneration Score

For Model 3, samples of one retina were fixed in one block but separated by the nitrocellulose membrane in freshly prepared 4% PFA (Applichem, Darmstadt, Germany) for 12 h at RT. Next, retinae were rinsed in PBS and dehydrated through a series of increasing EtOH concentrations and Xylol (Reactolab, Servion, Switzerland) (Table 2), before being embedded in Paraffin (Paraplast, Leica Biosystems, Muttenz, Switzerland) and stored at RT until use. Sections were analyzed for tissue integrity by bright field (BF) microscopy (Leica DM4000, Leica, Heerbrugg, Switzerland). One sample of the freshly isolated retina was mounted without fixation on a slide for BF microscopy.

For Models 5a and 5b, tissue processing was enhanced to preserve retinal laminar structure and to minimize damage during transfer from the culture insert to the microscope slide (Superfrost Plus, VWR). First, the slide was cleaned with EtOH and dried for 5 min before two small 5.5 mm strips of double-sided scotch tape were put on both ends of the slide. A 15 × 15 mm large coverslip (ThermoFisher Scientific) was placed on top of the scotch tape. Then, 1 mL sterile PBS was pipetted on the slide. The membrane of the culture insert was cut out with a scalpel no. 11 without touching the tissue and placed in a drop of PBS. Supported by forceps, the retina slipped in the PBS. The PBS was aspirated and replaced by a drop of mounting medium (ProLong Gold Antifade Reagent, ThermoFisher Scientific); a large cover slip (40 × 60 mm, ThermoFisher Scientific) was gently placed on the top and the sample was dried for 15 min. Samples were evaluated by BF microscopy at a magnification of 100× using the Leica DM4000. Five images were taken per sample with one micrograph from each peripheral side and one from the center. To quantify degeneration, a score from 0 to 3 was created, as shown in Table 3.

### 2.4. Hemalum and Eosin (H&E) Staining

After culture termination, samples were fixed in 4% PFA. For Model 1, every sample was fixed individually, for Models 2 to 4, all samples of one retina were fixed in one block, and for Model 5a/b, two samples were fixed and embedded in one block separated by nitrocellulose membrane. Except for Model 5a/b (2–4 h, RT), samples were fixed for 12 h at RT. In Model 5a/b, tissue preservation was optimized by placing samples in a cassette cushioned with filter paper on both sides and a mesh on one side to prevent the movement of the tissue in the cassette. After rinsing in PBS or distilled water, tissues were dehydrated via an increasing EtOH and Xylol series (Table 2). Then, samples were embedded in paraffin and cut into 5–6 μm thin slices. Before being stained with H&E (VWR), samples were deparaffined and rehydrated; finally, the tissue was mounted (Table 2). Two images per sample at 100×, 200×, and 400× magnification were recorded. 

### 2.5. Cell death and Viability Determination

#### 2.5.1. CytoTox Glo^TM^ Assay

**CytoTox Glo^TM^ assay Model 5a,b.** Retina viability was assessed by the CytoTox-Glo^TM^ assay (Promega, Fitchburg, MA, USA), which is based on the release of a protease by damaged cells with compromised membrane integrity. Measurements were performed in triplicate using the FLUOstar Omega spectrophotometer (BMG Labtech, Ortenberg, Germany) (settings: gain: 3600; positioning delay: 0.1; measurement start: 0.0; measurement interval time: 1.0; and time to normal: 0.0). Luminescence was directly used to compare cell viability at different days in standardized 6 mm punched samples and not normalized to a cell count. Supernatant was collected 24 h after medium was changed on days 3, 6, 9, 11, and 14. Fifty microliters of CytoTox-Glo reagent was added to 100 µL culture medium and incubated in the dark for 15 min at RT on an orbital shaker. A standard curve using ARPE-19 cells was prepared (10,000, 5000, 2500, 1250, 625, and 313 cells).

**CytoTox Glo^TM^ assay (after H_2_O_2_ treatment).** The same assay was used to determine viability of retinal tissue after H_2_O_2_-induced OS. Here, samples were weighed, and luminescence was correlated to the weight of the tissue. Retinae were cut into 2 halves using scalpel no. 10. With Colibri forceps, each half was transferred to a 1.5 mL tube (Milian) and weighed. Next, 50 μL of CytoTox-Glo reagent was added to each well, gently mixed, and incubated in the dark on an orbital shaker at RT for 15 min, before luminescence was measured. Subsequently, according to manufacturer’s instructions, lysis reagent was added, and luminescence was again measured after incubation in the dark. After 9 d of culture and the loss of the supportive nitrocellulose membrane, the tissue did not need a homogenization step; careful mixing with the assay reagent and shaking during incubation was sufficient.

#### 2.5.2. CellTiter Glo^®^ Viability Assay

The CellTiter-Glo^®^ cell assay (Promega) detects cellular ATP and thus determines the amount of metabolic active cells visualized by luminescence. Using a dermatological Biopsy Punch with a 5 mm diameter (Stiefel), size-standardized samples from the different times of culture and experimental conditions were prepared. Samples were homogenized by up- and down-pipetting in 500 μL of normal glucose medium. Ten microliters of one sample was diluted 1:1 in Trypan Blue (Fisher Scientific) and counted in a Neubauer chamber to estimate the cell number of all samples. Two hundred and fifty thousand cells in 100 μL of normal glucose medium per sample were incubated in a 96-well plate. The assay was performed in triplicates according to manufacturer’s instructions. Briefly, 100 μL CellTiter-Glo^®^ reagent was mixed with the cells in an orbital shaker for 2 min. The plate was incubated for 10 min at RT, and luminescence was recorded using the FLUOstar^®^ spectrophotometer; the gain was adjusted for each measurement, and an ATP standard curve of serial ten-fold dilutions (1 nM to 10 μM) was generated.

#### 2.5.3. Propidium Iodide (PI) Staining

Necrosis was determined in samples processed as described in Section 2.3 with duplicates per analysis day by PI staining. Forty microliters of the 2 mg/mL PI stock solution (Merck) was diluted in 4 mL sterile PBS to a final working concentration of 0.01%. To maintain retinae’s orientation, PI staining solution was added to the culture insert carefully from the side. Samples were incubated in the dark at 21 °C for 10 min. Next, tissues were rinsed twice with 1 mL of sterile PBS for 5 min before being mounted on a glass slide (Section 2.3), and microscopic pictures were immediately taken at 100× magnification using a fluorescence microscope. Five images were taken from every sample with one image from every peripheral side and one from the center.

#### 2.5.4. TUNEL Assay

The In Situ Cell Death Detection Kit^®^ Fluorescein (Merck) was used to determine apoptosis in paraffin-embedded retinal cross sections; broken DNA strands in apoptotic cells were detected by incorporation of fluorescein (FITC)-labelled nucleotides polymerized by the terminal deoxynucleotidyl transferase. Apoptotic cells were then visualized by fluorescence microscopy (Leica DM4000) and counted manually on images (1 micrograph/sample) of 200× magnification using ImageJ (software version 2.3.0/1.53q, National Institute of Health, Bethesda, MD, USA). The assay was performed according to the manufacturer’s instructions.

### 2.6. GSH Assay

The luminescent-based GSH-Glo™ assay (Promega) quantifies glutathione (GSH) in cells or tissues to assess cellular OS. Retinae were prepared as described in Section 2.5.1, and luminescence was correlated to the weight of the tissue. Each half was transferred to a 1.5 mL tube prefilled with 50 μL PBS. The assay was performed according to the manufacturer’s instructions. Briefly, 50 μL of the cell suspension was transferred to a 96-well plate, and 50 μL of GSH-Glo™ Reagent 2X was added to each well. After mixing the solution, it was incubated on an orbital shaker in the dark at RT for 30 min. Subsequently, 100 μL of the Luciferin Detection Reagent was added to each well, mixed, and incubated in the dark on an orbital shaker at RT for 15 min. Luminescence was measured using the FLUOstar Omega spectrophotometer, applying same settings as described in Section 2.5.1. A standard curve of 5 to 0.005 mM GSH was prepared.

### 2.7. Immunohistology

Single and double immunofluorescence staining were performed on paraffin-embedded retinal sections processed as described in Section 2.4. and counterstained with 4′,6-diamidino-2-phenylindole (DAPI) according to the protocol described in Table 4. Table 5 lists used primary and secondary antibodies. After mounting, micrographs were recorded at 100×, 200×, and 400× magnification using the Leica DM4000 fluorescence microscope, taking three images from every sample (right side, left side, center).

### 2.8. Statistics

Descriptive statistics were calculated for all data (mean, SD, SEM, min, max, range). A D’Agostino and Pearson normality test determined parametric and non-parametric data. Differences between groups were calculated using the parametric *t*-test (2 samples), ANOVA with Tukey’s multiple comparison test (≥3 samples; in case of missing values, a mixed effects model was applied), and non-parametric Mann–Whitney (2 samples) or Kruskal–Wallis with Dunn’s post-test (≥3 samples) depending on the nature of the data. Time courses were analyzed, applying a paired ANOVA. If not noted otherwise, data are shown as mean ± SD. All analyses were made using the GraphPad Prism^®^ software version 9.3.1 (471) (GraphPad Software, LLC, San Diego, CA, USA).

## 3. Results

### 3.1. Model 1: Static Bovine Retina-RPE Culture, 4 Days

Model 1 used bovine eyes and provided us with our first insight into the feasibility of ex vivo cultured retinae, especially when cultured in conjunction with choroid and RPE layers, and inflammatory reactions. The H&E-stained sections confirmed that the tissue integrity, including the photoreceptor outer segments (POS), was maintained for 4 d (Figure 2a). Nevertheless, signs of degeneration were visible, i.e., holes and cell loss in the retinal ganglion cell (RGC) and inner nuclear (INL) layers, and the connection between the RPE and POS got lost. The protein kinase C (PKC) selectively stains bipolar cells, indicating cell loss (reduced staining) or inflammation (increased staining); in the present model, the PKC expression (green) did not decline over time (Figure 2b, bottom) compared to the sections stained directly after isolation (Figure 2b, top). The second inflammatory marker, the glial fibrillary acidic protein (GFAP), identifies “activated” Müller cells; the resulting pictures demonstrated a low, positive GFAP signal on day 4 of culture (Figure 2c, bottom) comparable to that of retinae stained directly after isolation (Figure 2c, top).

### 3.2. Model 2: Dynamic Porcine Retina Culture, 3 Days

Model 2 cultured porcine retinae and tested a perfusion culture. The aim of the study was to evaluate the potential advantages of a perfusion culture to optimize media and to compare the benefit of isolating the RPE–retina complex vs. using the neural retina alone. To that goal, the cell counts of the retinal layers performed in H&E-stained retinal cross sections from retinae cultured for 1 d with or without RPE were compared to those of retinae stained directly after isolation (“fresh”). The cultured neural retinae showed comparable cell counts as to those of the directly processed tissue, while the cell counts were significantly reduced in the RPE–retina cultures in the outer nuclear (ONL), INL, and RGC layers (Figure 3a,b). The H&E-stained cross sections shown in Figure 3c confirmed the samples isolated and cultured without RPE had a better-preserved structure compared to that of the RPE–retinae cultures.

Figure 4 reports the results of a comparison of the Ames medium and DMEM in the neural retinae cultured for 1 d. In the ONL, the cell counts did not differ significantly but showed a trend of lowered cell numbers in DMEM (Figure 4a,b). In the INL and RGC layers, this difference was significant, with reduced cell counts after being cultured in DMEM. This result is supported by better tissue preservation, as shown in the H&E-stained cross sections from the retinae cultured in Ames medium (Figure 4c).

Next, culture temperatures of 37 °C vs. 21 °C were compared. The cell counts in the ONL decreased from “21 °C” to “fresh” and “37 °C” samples. Additionally, in the INL, the highest numbers were counted in the “21 °C” group, followed by almost similar counts in the “fresh” and “37 °C” groups (Figure 5a,d). In the RGC, no differences between the groups were detected. Generally, the differences were not significant. However, a TUNEL assay revealed significant differences in apoptosis due to culture temperature (Figure 5b,d). The number of apoptotic cells was significantly lowered after the culture at 21 °C compared to the 37 °C cultures, even compared to freshly prepared sections. Representative H&E-stained sections confirmed overall increased degeneration in retinae cultured at 37 °C, as demonstrated by large holes and cell loss. Nevertheless, holes were also found in the inner plexiform (IPL) and nerve fiber (NFL) layers of the retinae cultured at 21 °C (Figure 5c).

### 3.3. Model 3: Static Porcine Retina Culture, 4 Days

The culture conditions established in Model 2 were adapted to the static retina culture in Model 3. Figure 6a–e shows representative images of different stages of degeneration at different time points, which are quantified in Figure 6f. Degeneration increased over time but stayed on average below the highest score of ‘3’. The mean score was 0.80 ± 0.45 on day 1, which increased to 2.0 ± 0.82 on day 2, 2.13 ± 0.64 on day 3, and 2.33 ± 0.82 on day 4; the differences were significant for 1 vs. 3 d (*p* = 0.0496) and for 1 vs. 4 d (*p* = 0.0234).

### 3.4. Model 4: Static Rat Retina Culture, 9–13 Days

The optimization of culture conditions could significantly slow down degeneration in Models 2 and 3; nevertheless, the culture duration was still short. Thus, in Model 4, the effect of using fresh tissue isolated immediately after the animals’ sacrifice was evaluated. Tissue preservation was additionally improved by supplementing the media with growth factors N-2 and B-27. The use of fresh tissue enabled a robust culture prolongation until 9 d. The retinae were cultured up to 13 d; however, after 9 d, degeneration became prominent, often not allowing us to perform a reliable tissue analysis. The H&E-stained sections demonstrated tissue preservation from days 1 to 9 and a non-significant, acceptable number of holes and cell loss (Figure 7a–d). The POS were partly preserved until 9 d. At later time points, in many samples, either the thickening of the tissue pointed to inflammatory reactions (Figure 7e) or the laminar structure of the retinae was lost. Moreover, the INL and ONL were no longer distinguishable, and the thinning of the tissue and cell loss were observed (Figure 7f).

### 3.5. Model 5 and 5b: Static Porcine and Human Retina Culture, 14 Days

Knowledge from Model 4 was applied to porcine eyes. A significant improvement in tissue isolation and processing also enabled us to transfer the model to the retina culture from human donor eyes cultured 6–8 d post mortem. It was possible to preserve the laminar structure and distinguishable retinal layers for 14 d in both the porcine and human retinae, as shown in the H&E-stained tissue sections (Figure 8a,b). From day 1–14, the main retinal layers (RGC, IPL, INL, OPL, ONL) were distinguishable and did not show significant thinning (indicating cell loss) or thickening (indicating inflammation). The POS were hardly visible in the H&E-stained sections. The retinal thickness was determined in micrographs from the H&E-stained sections (Figure 8c). The porcine samples had a thickness from 54.74 ± 12.81 µm (day 1) to 65.59 ± 11.60 µm (day 6) and 46.42 ± 10.48 µm (day 14) without significant differences (*p* = 0.4882); the thickness increased slightly on day 6 of the culture but decreased back to the base values at day 14. Moreover, the thickness in the human retinae did not differ significantly over time (*p* = 0.6126), with 51.09 ± 15.28 µm (day 1) to 66.58 ± 19.16 µm (day 6) and 53.08 ± 3.08 µm (day 14).

Tissue quality was also analyzed by determining the degree of degeneration (score), viability (CytoTox-Glow^TM^), and necrosis (PI staining) (Figure 9). The cell viability correlated inversely to the measured luminescence; thus, the lower the luminescence, the higher the viability. The black curve (pRetina) illustrated a high luminescence on day 3 which continuously decreased over time, indicating a decrease in cell death (Figure 9a). The viability in hRetina remained high over the whole culture period (Figure 9a, grey curve). The amount of necrosis only slightly increased over time for both species (Figure 9b). Figure 9c shows the degeneration score over 14 d for pRetina; the score increased over time but generally remained low with a maximum score of 2.13. The degree of degeneration in hRetinae was even lower, with a maximal score of 1.60 at 14 d (Figure 9d).

Through immunohistochemical staining, we analyzed the cell loss using cell-type-specific antibodies (rhodopsin for POS, vimentin for Müller cells) and detected potential inflammatory reactions using anti-GFAP and anti-Iba-1 antibodies. Vimentin staining (red) confirmed normal Müller cell morphology for the whole culture period (Figure 10g–l). Rhodopsin staining (green) detected the POS for 14 d in both species. GFAP staining (red) increased on day 6 in pRetinae and hRetinae but decreased even below the base expression at day 14 (Figure 10a–f). Iba-1 staining (green) remained low at every time point in both species (Figure 10g–l).

### 3.6. Antioxidant Scutellarin to Treat OS in High-Glucose Retina

Models 3 and 4 were used to analyze disease conditions linked to OS, i.e., high-glucose conditions and incubation with H_2_O_2_ were used to test antioxidative treatment options. pRetinae (Model 3) were cultured in normal glucose (NG, 6 mM D-glucose) and high-glucose (HG, 30 mM D-glucose) conditions to provoke cellular damage, as is seen in patients suffering from DR. Scutellarin was added, and its effect was tested in HG conditions. The cell viability was determined by the cells’ capability to produce ATP in the retinae cultured in HG and with 0, 1, 10, or 100 µM scutellarin added to the cell culture medium. At 24 h, the viability increased after the medium was supplemented with 1 µM (998.9 ± 202.5 nM ATP) and 10 µM scutellarin (783.6 ± 164.6 nM ATP), while the addition of 100 µM scutellarin did not increase the viability (360.1 ± 152.8 nM ATP) compared to that of the NG (500.3 ± 17.01 nM ATP) and HG (560.1 ± 138.7 nM ATP) conditions (Figure 11a). The viability after the treatment with 1 µM scutellarin differed significantly to that of the treatment with 100 µM scutellarin (*p* = 0.0053). After 4 d of culture at HG conditions (Figure 11b), its detrimental effect became more obvious from the decreased ATP level of 288.3 ± 59.9 nM compared to 435.7 ± 166.8 nM ATP in NG conditions. Scutellarin (1 µM: 515.9 ± 172.0 nM ATP; 10 µM: 420.5 ± 201.9 nM ATP) increased the viability on day 4 to levels comparable to those of the NG conditions; meanwhile, 100 µM scutellarin did not increase the viability (325.7 ± 154.7 nM ATP). The antioxidative effect of low and medium doses of scutellarin was illustrated in the H&E-stained cross sections (Figure 11c); only after the treatment with 1–10 µM scutellarin were the POS preserved. Nevertheless, degeneration, i.e., holes and cell loss, was still visible.

Apoptosis was determined on days 1 and 4 (Figure 12). On day 1, apoptosis was highest in the HG conditions (69.60 ± 27.70 cells/image) and decreased even below that of the NG conditions (46.2 ± 24.41 cells/image) after the addition of scutellarin. The beneficial effect was similar for all doses: 1 µM = 32.25 ± 7.50; 10 µM = 41.60 ± 18.06; and 100 µM = 37.75 ± 16.09 cells/image. The difference between the freshly fixed control and the HG conditions was significant (*p* = 0.0050). On day 4, apoptosis generally increased with similar differences between the groups. The HG conditions had the highest values (124.20 ± 51.09 cells/image), followed by the NG conditions (71.83 ± 35.38 cells/image). Apoptosis was further decreased by the scutellarin to 64.80 ± 30.68 cells/image (1 µM), 75.20 ± 321.19 cells/image (10 µM), and 60.80 ± 319.66 cells/image (100 µM). The difference between the fresh control (2.75 ± 3.40 cells/image) and HG groups was significant (*p* = 0.0017). Micrographs were created to visualize the results on day 4 (Figure 12c), showing that the few apoptotic cells seen in the NG group were located in the ONL, while the majority of the apoptotic cells detected in the HG group were found in the INL. After the treatment with scutellarin, apoptosis decreased in all the tested groups; however, after the treatment with 1 µM (HG1) and 100 µM (HG100) scutellarin, the remaining apoptotic cells were mainly found in the ONL, while after the treatment with 10 µM (HG10) scutellarin, the remaining apoptotic cells were mainly found in the INL.

### 3.7. OS Reduction and Cell Protection by PEDF and GM-CSF in H_2_O_2_-Treated Retina

OS induced by age-related cellular alterations plays a key role in aAMD pathogenesis. Thus, a reduction in OS might slow down or halt retinal degeneration. For this purpose, RPE or IPE cells were transfected using the SB100X transposon system with the genes coding for the PEDF and GM-CSF proteins. To test the proteins’ cell-protective effect, rRetinae cultured for up to 13 d (Model 4) were treated with one or both proteins for the first 3 d of culture; on day 3, OS was induced by H_2_O_2_. The OS levels of the treated retinae were determined by a GSH assay; cell-protective effects were quantified by a viability assay (Figure 13a,c). After the H_2_O_2_ treatment, the tissue was extremely fragile, such that not all of the measured samples delivered technically valid results. Therefore, the rate of valid/invalid results was analyzed as an additional parameter for OS and the antioxidative effect of PEDF/GM-CSF, respectively (Figure 13b,d). The cell viability (Figure 13a) in the freshly fixed retinae (control) was 31.03 ± 19.50%. In contrast, the tissue incubated with H_2_O_2_ was highly damaged, and no valid results could be measured, indicating a low viability. The treatment with PEDF and GM-CSF increased the viability significantly to 45.44 ± 9.02% (PEDF) and 50.40 ± 7.65% (GM-CSF), when “normal” tissue (no H_2_O_2_ incubation) was treated with the proteins. After the H_2_O_2_ incubation, viabilities of 52.80 ± 18.24% (PH = PEDF + H_2_O_2_) and 60.07 ± 13.26% (PGH = PEDF + GM-CSF + H_2_O_2_) were measured. No valid results were measured after the treatment with GM-CSF only (GH = GM-CSF + H_2_O_2_). The highest percentages of valid viability measurements were seen in the non-H_2_O_2_-incubated groups with 100% (fresh), 100% (PEDF), and 71% (GM-CSF) (Figure 13b). No valid results (0%) could be attained in the H_2_O_2_-incubated and GH-treated samples. The treatment with PEDF or PEDF + GM-CSF increased the number of valid results to 25% (PH) and 38% (PGH). The lowest GSH levels were measured in the samples that were not incubated with H_2_O_2_ (fresh: 0.01 ± 0.00 µg/10 mg tissue; PEDF: 0.01 ± 0.00 µg/10 mg tissue; GM-CSF: 0.02 ± 0.00 µg/10 mg tissue) (Figure 13c). The GSH level increased after the H_2_O_2_ treatment to 0.16 ± 0.00 µg/10 mg tissue; when additionally treated with PEDF and/or GM-CSF, the levels further rose (except the PH group with 0.13 ± 0.07 µg/10 mg tissue) to 0.23 ± 0.19 µg/10 mg tissue (GH) and 0.221 ± 0.30 µg/10 mg tissue (PGH). Tissue protection, i.e., lowered OS, was reflected by the percentages of valid GSH results in the GM-CSF- (83.3%), and PEDF-GM-CSF-treated retinae (100%) after H_2_O_2_ incubation (Figure 13d). If treated with PEDF alone, the percentage of valid results did not differ to that of the H_2_O_2_-incubated, untreated retinae (66.7%). The lowest numbers of valid results were seen for the “normal” tissue: one of three for the fresh tissue and one of five for the tissues treated with PEDF and GM-CSF, respectively.

## 4. Discussion

ROS, which are continuously produced especially by mitochondria [45,46], are important signaling molecules with beneficial effects induced by calorie limitation, glucose limitation, and physical exercise (referring to the mitohormesis concept) [45]. However, an excess of ROS induced by mitochondrial dysfunction and a downregulation of antioxidant enzymes leads to OS and cell damage, and plays a key role in normal aging and age-related diseases [45,46]. ROS such as the superoxide anion (O_2_^•−^), H_2_O_2_, or the hydroxyl ion (OH^−^) damage proteins, nucleic acids, and lipids, leading to cancerogenesis, cellular dysfunction, and death [46]. The body developed a three-level antioxidative defense system against cell damage. First, superoxide dismutases (SODs) and related enzymes remove H_2_O_2_ and lipid hydroperoxides. Secondly, thioredoxin, glutamate–cysteine ligase, and the glutathione synthetase generate GSH, glutathione, and thioredoxin reductases to reduce glutathione and thioredoxin disulfide. Finally, the NF-κB pathway and others are activated to eliminate oxidized macromolecules, repair damage, and inhibit apoptosis, e.g., by regulating the expression of Bcl-2 proteins [45,46]. In ageing, an increased ROS level and decreased antioxidant defense results in the frailty syndrome, characterized by a reduced physiological reserve and increased susceptibility to stressors; when this weakened balance is disrupted, age-related diseases evolve [3].

AMD is a typical age-related disease, and as long-term complication of diabetes, DR can also be numbered among this group; untreated, both lead to retinal neurodegeneration and blindness [5]. As an organ with particularly high levels of oxygen consumption and light processing to generate vision, the retina is constantly producing ROS and is thus particularly prone to OS [47]. A key process in the pathogenesis of aAMD is the interplay of inflammation and OS that leads to the dysfunction of the RPE. Mitochondrial damage, ATP depletion, inflammation, and Receptor-interacting serine/threonine-protein kinase 3 (RIPK-3)-mediated membrane damage lead to apoptotic and necrotic cell death [47]. Oxidatively modified lipids, 89 proteins including the peroxiredoxin 1 enzyme from the antioxidant defense system, and trace elements accumulate at the interface of Bruch’s membrane and the RPE and form drusen [48,49]. As a consequence, the RPE cannot anymore support the functioning and survival of photoreceptors, which also undergo cell death [11,47]. The diabetic retina is harmed by lipid peroxidation, protein oxidation, and DNA damage, and endogenous levels of antioxidant defense enzymes are diminished; meanwhile, pro-apoptotic proteins are increased, and levels of neurotrophic factors are decreased [50]. The prevalence of AMD and DR is increasing due to aging populations and increasing rates of diabetes [51] and they will become a severe burden for patients and health care systems [6]. In industrialized countries, they are already the first cause of blindness in elderly patients (AMD) and the working-age population (DR) [10]. Thus, there is an urgent need for innovative, reliable models for a better understanding of the pathogenesis of OS-triggered diseases and the development of novel therapies. Such models are ideally (i) simple to generate and thus easily reproducible, (ii) available at a low cost, (iii) able to be up-scaled, (iv) ethically favorable, (v) biologically complex, and (vi) transferable to patients.

Cell culture systems fulfill many of these criteria; however, they cannot evaluate complex biological interactions and are limited in their transferability to patients. The Ala Octa^®^ affair demonstrated issues with in vitro studies: Perfluorooctane (Ala Octa^®^) was a perfluorocarbon liquid used in retinal surgery as transient tamponade and its cytotoxicity was tested—accordingly to regulatory standards—on mouse fibroblasts. Pastor et al. could demonstrate that as a result of insufficient testing, its cytotoxicity was underestimated and led to severe ocular complications and even blindness in many patients [15]. In vivo studies are more suitable to detect potential local and systemic adverse effects. However, research institutions, the pharmacological industry, regulatory and governmental bodies are motivated to reduce animal experiments in biomedical research to a minimum following the 4R principles [14]. Moreover, many animal models are only somewhat transferable to patients. Pennesi et al. reviewed AMD animal models [52], most of which are working with mice; these do not have a macula and are active at night with a very different retinal anatomy and cellular composition compared to those of a human. Often, the animals are of a young age [52] and are useful to investigate distinct disease features, but insufficient in proving the efficiency and safety of novel treatments against age-related maculopathies. Studies with larger animals and species that have a macula are possible; however, the larger the animal, the higher the costs and the more ethical questions arise. Organ cultures offer an alternative. They are relatively easy to generate, available at low costs compared to in vivo studies, allow for the study of the intercellular interactions of complex tissues, and are ethically favorable. Moreover, the use of human donor tissue promises transferability to patients and enables us to evaluate diseased or aged organs. The retina culture was first reported by Strangeways and Fells [17] who analyzed the ocular development of cultured embryonic chick eyes for 32 d. The culture of adult retinae is more challenging: recent research has reported a suspension culture of retinal cells for 8 weeks [18], but, except for single reports, an organotypic culture with a preserved tissue structure can be maintained for 7 to 14 d [19,53]. Thus, the retina culture is an excellent system to analyze organ development, disease pathogenesis, intoxications, novel drug candidates, and treatment approaches, particularly for OS, but further studies are needed [19,53].

In the present study, five retina culture models comparing different isolation and tissue processing methods, media, temperature, mammalian species, and post mortem times were presented (as summarized in Table 1). Briefly, the use of porcine or bovine eyes provided by the food industry is ethically preferable to laboratory rat tissue. By applying enhanced isolation and tissue processing techniques, cultures of retinae with a prolonged post mortem time are not inferior to fresh rat tissue in terms of tissue morphology, degeneration, cell viability, and death. The Ames culture medium, optimized by supplementation with growth factors N-2 and B-27 and retina incubation at 21 °C, revealed superior culture conditions. The implementation of an enhanced isolation and tissue processing technique (“minimal-touch technique”) allowed for the transfer of the culture system from animal to human donor retinae, which are most fragile due to having a post mortem time of several days. The culture of the neural retina together with the choroid and/or RPE cell layers seemed to be cell-protective; however, the complex isolation techniques necessary offset this benefit.

Various species are used for retina cultures, of which pig and cattle are frequently used since they are easily available. Kuehn et al. evaluated hypoxia-induced retinal degeneration following cobalt-chloride treatment in porcine retinae (8 d culture; Neurobasal medium supplemented with glutamine, B-27, and N-2; photoreceptors facing the multi-well inserts with 4–6 samples/retina; isolation using a 6 mm punch) [25]. Their general tissue quality was very good, enabling the analysis of the dose-dependent effects of the cobalt-chloride treatment. Similarly, Models 2, 3, and 5a used porcine tissue and confirmed excellent tissue quality, which enabled a retina culture up to 14 d (Model 5a). Model 1 used cattle as the starting material, as did Peynshaert et al. (2 d culture; isolation and culture of neural retina and vitreous, Neurobasal medium supplemented with glutamine and B-27; three explants/eye,) [23]. The results demonstrated excellent tissue quality in terms of tissue architecture, cell composition, and the nerve fiber layer and were functional comparable to the data from Model 1. Schnichels et al. isolated young rat retinae immediately after sacrifice (7 d culture; R16-complete medium; retina of one eye cultured as one sample) [54]. The retinal architecture was maintained for 7 d, and the tissue thickness decreased only moderately. In line with these data, Model 4 demonstrated a retarded degeneration and prolonged culture duration of up to 9–13 d when using freshly isolated rat retinae (two samples/eye). It is assumed that the short post mortem time and not the species was responsible for the positive influence on cell viability in both studies. Yet, since Model 4 requested the sacrifice of one animal for four samples, it is ethically questionable and not scalable.

The perfusion culture presented in Model 2 reported first by Kobuch et al. [55] guaranteed constant nutrient and oxygen delivery. The culture of the choroid–RPE–retina complex in a perfusion chamber allowed for tissue preservation up to 10 d with the POS showing the first signs of degeneration after 1 d. Apoptotic and necrotic cell death was seen after 4 d, while in the control samples (static culture), the tissue structure was disrupted after 4 d (10 d culture; DMEM, 15% porcine serum, 2.5% HEPES buffer; one sample/retina) [55]. Similarly, Model 2 reported a decelerated degeneration compared to that of Model 1; however, compared to Models 3 and 5, no significant advantage could be observed; it is assumed that the daily medium change (in static culture) delivered sufficient nutrients and oxygen for optimal tissue preservation. Thus, considering the simpler setup and easier up-scaling of static cultures, this system seems preferable.

Models 1 and 2 cultured the neural retina with supportive choroid and/or RPE tissue, since Kobuch et al. reported a benefit of this technique on retina conservation [55]. Additionally, Peynshaert et al. reported low levels of inflammation, no significant changes in tissue thickness, and stable populations of bipolar cells after a culture of the neural retina with supportive tissue (vitreous) [23]. Indeed, the POS were especially better preserved when cultured together with RPE cells (and the choroid); however, enlarged analyses performed with Model 2 did not confirm the prominent differences between the culture of the neural retina alone or with (choroid-)RPE tissue. The higher complexity of the isolation technique seemed to negate the positive effect of the supportive tissue layers, i.e., not all samples could be put in culture or showed damage due to isolation.

It is known that hypothermia causes immunosuppression and slows down metabolism and, thus, protects organs from damage due to hypoxia, ischemia, or trauma. This was previously shown by Narayanamurthy et al., who presented a cooling device that protected the rat brain from hypoxic-ischemic encephalopathy [56]. A hypothermia-induced metabolism deceleration significantly inhibited the development of OS and its consequences, as shown by Choi et al. in a model of renal ischemia-reperfusion injury [57]. Moreover, various ocular conditions, such as allergic conjunctivitis or retinal hypoxia and ischemia, are treated with hypothermia [58,59]. Due to the high metabolic demand and metabolic turnover of the retina, the idea to rescue cultured retinal tissue by hypothermia was evaluated by Ames and Gurian in 1963: they showed that rabbit retinae recovered more rapidly at 30 °C than at 37 °C after glucose and oxygen deprivation [60]. Recently, Mueller-Buehl et al. reported the OS-reducing effect of hypothermia on H_2_O_2_-induced damage in porcine RGCs [28]. Accordingly, Model 2 cultured retinae in hypothermic conditions (21 °C) compared to samples cultured at 37 °C and revealed significant tissue preservation by higher cell counts and a lowered rate of apoptosis. It has to be noted that pharmacokinetics analyses have to be analyzed carefully due to the slowed metabolism.

While in Model 1, the retina samples were cultured in DMEM, in Model 2, Ames medium was introduced and used in all of the following systems. The results of Model 2 demonstrated a significant benefit of Ames medium especially developed for the culture of retinal tissue [61]. Many other groups are using Neurobasal medium for its protective function for neurons and have reported positive results [23,25]. However, particularly Model 5 confirmed that a medium that imitates the fluid that “bathes the retina in vivo” [61] seems ideal to support this complex tissue, which is composed of multiple neural and non-neural cell types. It has to be noted that in Model 4, the medium has been optimized by the addition of growth factors B-27 and N2 among other factors, enabling a two-week-long culture.

The experimental series in high-glucose conditions and after H_2_O_2_ treatment demonstrated the benefit of the retina culture models in preclinical testing. Model 3 cultured porcine retinae in a static culture using Ames medium at 21 °C. Tissue degeneration could be kept at a low level for 4 d during which the detrimental effect of OS induced by “high glucose” and the benefit of treatment with the antioxidant scutellarin could be determined. It has to be noted that high doses of scutellarin might negate its positive effects, but more detailed analyses have to be conducted to confirm this tendency. Model 4 has been implemented to prolong the culture, and its avail was evaluated by analyzing the effect of PEDF and/or GM-CSF on cell viability and OS after the H_2_O_2_ treatment. The rat retinae were cultured immediately after their sacrifice, which allowed cultures up to 13 d and confirmed the cell viability rates in freshly fixed retinae to be around 31%, which has also been measured in IPE cells immediately after isolation [62]. After the H_2_O_2_ treatment, reduced GSH levels were expected as reported by Wang et al. in ARPE-19 cells [63] and by Zheng et al. in retinae from 18-month-old hydroquinone-fed mice [64]. In contrast, the results showed the lowest GSH concentrations were in the controls, while the GSH levels in the H_2_O_2_-treated samples were increased. It is assumed that while already improved compared to Models 1–3, the tissue quality in Model 4 was not good enough to reliably measure the GSH levels. This is supported by the low percentage of samples that delivered methodologically valid results.

Thus, Model 5 was developed to (i) generally increase tissue quality and prolong culture duration, (ii) implement a procedure that allows for the up-scaling of the testing series, and (iii) enable transference to human donor tissue with a longer post mortem time. Porcine retinae were cultured in static conditions at 21 °C in Ames medium supplemented with growth factors. Moreover, the isolation and tissue processing techniques were improved. Our novel “minimal-touch technique” included sample preparation by 6 mm punches up to 32 samples/retina, as well as special tissue handling and processing as detailed in Section 2. It enabled significantly improved retina preservation shown by low degeneration scores on day 14, high levels of viability, low apoptosis, and low inflammation. The results allowed transference to human retinae with similar results and were comparable to data reported by other groups, such as Jüttner et al. [65] and Schnichels et al. [66]. Ongoing experiments are repeating the analyses of Model 4, evaluating the protective effect of PEDF and GM-CSF in OS to confirm the superiority of Model 5. Future enhancements will implement a co-culture of the neural retina with iPS-derived RPE cells to keep the simplicity of the model while adding supportive RPE tissue. Electroretinogram examinations will be added to the panel of analyses, and OS-related damage will be evaluated in more detail by analyzing SODs, lipid peroxidation products, and DNA oxidation products. This will be of particular interest when using human donor eyes from AMD or DR patients.

## 5. Conclusions

Five retina organ culture models from different mammalian species, including from humans, were evaluated for the preclinical testing of OS, and antioxidant intraocular therapy approaches were presented. The quality of starting materials, tissue handling techniques, and culture conditions including hypothermia were assessed. The static culture of the neural retina in Ames medium on transwell inserts with the photoreceptors facing the insert at 21 °C resulted in optimal retina preservation, even in human donor retina samples with a post mortem time of up to 8 d (Model 5b). Further improvements are in development to allow for more detailed analyses of OS-related damage. At the present, we show that the combination of Models 5a (porcine) and b (human) enables a two-week-long retina culture in large experimental series for a better understanding of the pathogenesis of age-related, OS-triggered retinopathies and for extended drug candidate testing.

## Figures and Tables

**Figure 1 antioxidants-12-01211-f001:**
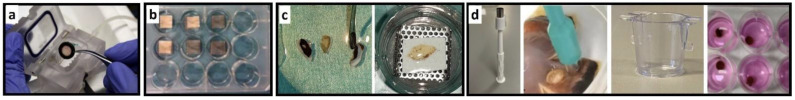
Key features of presented retina culture models. The photographs illustrate the different culture systems. (**a**) Minuth chamber with a retina clamped into the culture ring. (**b**) In the multi-well plate are pRetinae cultured on nitrocellulose membranes on customized stainless-steel grids. (**c**) The smaller rat retina is cut into halves before cultured on nitrocellulose membranes and customized grids. (**d**) The panel shows the punch, how it cuts tissue, and the culture insert before use and in use.

**Figure 2 antioxidants-12-01211-f002:**
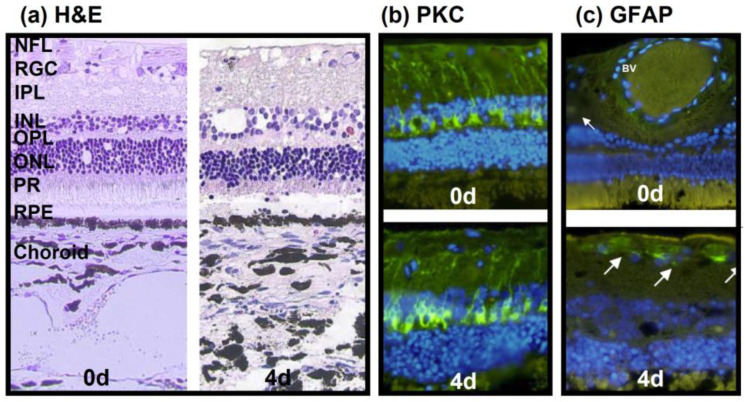
Morphological, functional, and inflammatory status of bovine retinae cultured with choroid and RPE. (**a**) H&E-stained sections of the choroid–RPE–retina complex processed directly after isolation (0 d, **left**) and on day 4 in culture (**right**), both showed good tissue preservation. INL and RGC layers showed reduced cell numbers and holes on day 4 in culture. (**b**) PKC expression in bipolar cells (green) was stable during ex vivo culture (**bottom**, 4 d) compared to sections stained directly after isolation (**top**, 0 d). (**c**) GFAP expression remained low in retinae on day 4 in culture (**bottom**, 4 d) comparable to that of samples stained directly after isolation (**top**, 0 d). White arrows indicate areas of weak GFAP expression. Nuclei are counterstained using DAPI. NFL: nerve fiber layer; RGC: retinal ganglion cells; IPL: inner plexiform layer; INL: inner nuclear layer; OPL: outer plexiform layer; ONL: outer nuclear layer; PR: photoreceptors; RPE: retinal pigment epithelium; BV: blood vessel.

**Figure 3 antioxidants-12-01211-f003:**
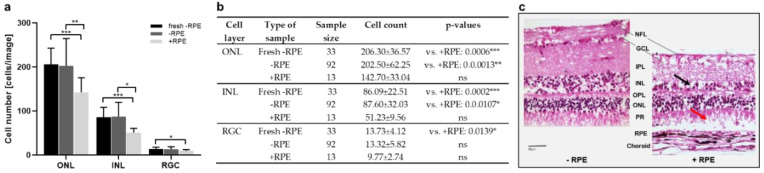
Cell counts and morphology of porcine retinae cultured with and without RPE. (**a**) Illustrated are cell counts in the ONL, INL, and RGC layers, determined in H&E-stained sections on day 1 from tissue samples isolated with (+RPE) and without (−RPE) RPE, compared to those of the sections directly stained after isolation without RPE (fresh −RPE). While cell counts of cultured, neural retinae were similar to freshly processed tissue, cell numbers were significantly reduced in ONL, INL, and RGC layers of cultured RPE–retina complexes. (**b**) Individual values, n-numbers, and *p*-values of cell counts in ONL, INL, and RGC layers. (**c**) Representative H&E-stained sections illustrate tissue degeneration in retinae cultured with/without RPE. Overall, the “+RPE” culture decreased in thickness and revealed holes and cell loss, as indicated by the black arrow in the INL and the red arrow in the POS layer. Magnification: 400×; scale bar: 50 µm. * = *p* < 0.05; ** = *p* < 0.01; *** = *p* < 0.001.

**Figure 4 antioxidants-12-01211-f004:**
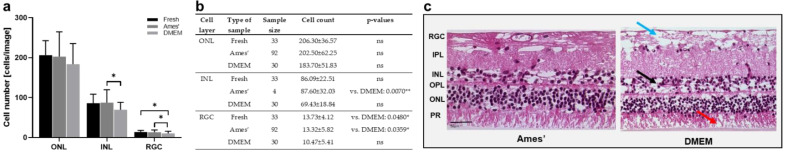
Cell counts and morphology of porcine retinae cultured in Ames medium or DMEM. (**a**) Illustrated are cell counts in the ONL, INL, and RGC layers, counted in H&E-stained sections on day 1. The cell counts showed slightly reduced cell numbers in the ONL of tissue cultured in DMEM; in the INL and RGC layers, this reduction was significant. (**b**) Individual values, n-numbers, and *p*-values of cell counts in ONL, INL, and RGC layers after culture in Ames medium vs. DMEM. (**c**) H&E-stained sections illustrate representative retinae cultured for 1 d in Ames or DMEM medium. Cross sections demonstrated signs of degeneration in DMEM-cultured retinae with large holes in the RGC (blue arrow), holes and cell loss in the INL (black arrow), and less structured, degenerated POS (red arrow). Fresh = stained directly after isolation. Magnification: 400×; scale bar: 50 µm. * = *p* < 0.05.

**Figure 5 antioxidants-12-01211-f005:**
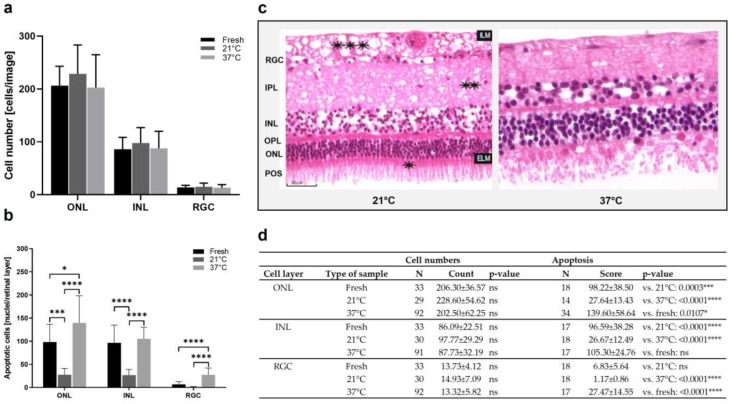
Cell counts and apoptosis in porcine retinae cultured at 37 °C vs. 21 °C. (**a**) Increased cell counts were detected in the ONL of retinae cultured at 21 °C for 1 d. Additionally, in the INL, highest cell counts were found in 21 °C cultures. There was no important difference in cell counts in the RGC. (**b**) The rate of apoptosis declined significantly in ONL, INL, and RGC layers of cultures incubated at 21 °C. (**c**) H&E-stained sections show representative examples of retinae cultured at 37 °C or 21 °C. Significant degeneration was seen after culture at 37 °C with holes mainly in the INL and a significant loss of tissue integrity and POS. Additionally, at 21 °C, holes were found in the NFL and IPL (asterisks) layers. (**d**) Individual values, n-numbers, and *p*-values of cell count and apoptosis analysis in ONL, INL, and RGC layers. ILM: Internal limiting membrane; ELM: external limiting membrane. Magnification: 400×; scale bar: 50 µm. * = *p* < 0.05; *** = *p* < 0.001; **** = *p* < 0.0001.

**Figure 6 antioxidants-12-01211-f006:**
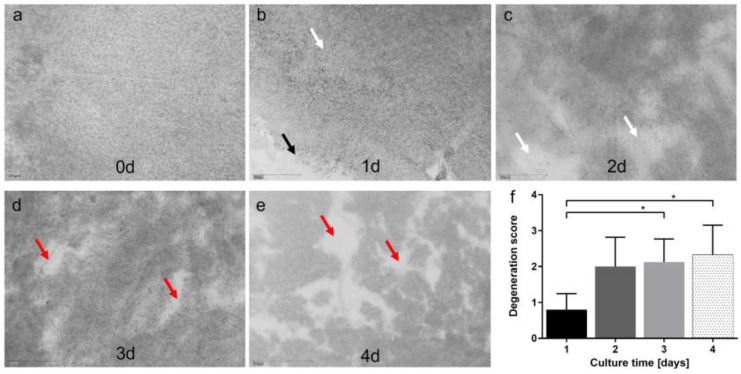
Degeneration score in static porcine retina culture for 4 d. (**a**) After isolation, retinae showed an integer structure without holes or clefts. (**b**) On day 1 of culture, the tissue was still of good quality, but for the brighter areas we can assume cell loss occurred (white arrow). Tissue borders were particularly exposed to degeneration (black arrow). (**c**) On day 2, degeneration became significant by bright spots, demonstrating lower cell density, i.e., cell loss (white arrows). (**d**) Degeneration further increased on day 3 and bright areas progressed to holes (red arrows). (**e**) After 4 d of culture, multiple holes and clefts demonstrated progressed tissue degeneration (red arrows); nevertheless, it must be noted that tissue assembly was still intact. (**f**) Degeneration was quantified by the degeneration score. Though degeneration progressed over time, it never reached the highest score of ‘3’, i.e., a destruction of the tissue assembly. Significant differences were found for 1 vs. 3 and 1 vs. 4 d of culture (*p* = 0.0496 and *p* = 0.0234, respectively). N = 5–8. Magnification: 200×; scale bar: 200 µm. * = *p* < 0.05. [37].

**Figure 7 antioxidants-12-01211-f007:**
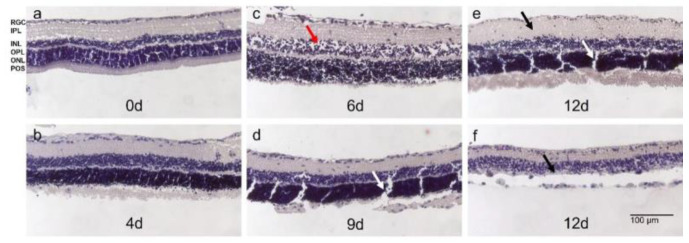
Morphology of H&E-stained rat retinae in static 12-day-long culture. (**a**) Retinae stained directly after isolation showed the typical laminar structure without signs of degeneration; POS were present. (**b**) On day 4, tissue quality was comparable to directly stained retinae without significant signs of degeneration and preserved laminar structure. (**c**) POS showed signs of degeneration after 6 d of culture. However, other retinal layers were preserved despite some holes indicating cell loss (red arrow). (**d**) Until day 9 of culture, tissue integrity was maintained without significant signs of degeneration except for the loss of POS. (**e**,**f**) At 12 d of culture, degeneration became significant. Either retinae thickened indicating inflammatory processes (black arrow €(**e**)) or cell loss was significant and retinal layers, particularly INL and ONL, were no longer distinguishable (**f**). Clefts in (**d**,**e**) (white arrows) were due to tissue processing. N = 3. Magnification: 100×; scale bar: 100 µm [43].

**Figure 8 antioxidants-12-01211-f008:**
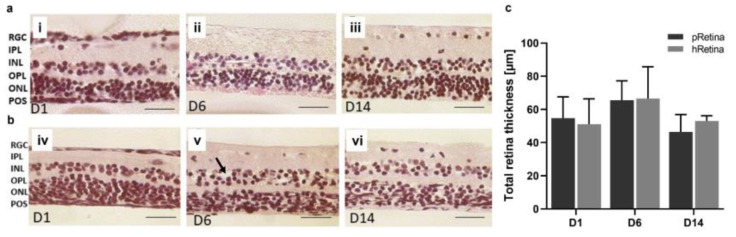
Morphology and thickness of H&E-stained porcine and human retinae after static culture for 14 d. Sample isolation and tissue processing were improved (Section 2.2.5) allowing for a 14-day-long culture. Shown are H&E-stained retinal sections from day 1 to day 14. (**a**) The presented sections illustrate the quality of retinae at the time point of isolation, at 6 and 14 d. On day 1, layers were distinguishable, no signs of degeneration were seen (**i**). Moreover, on day 6, retinae look normal, though little cell loss might have occurred in the INL and ONL (**ii**). POS were present. After 14 d of culture, POS were no longer visible in H&E-stained sections, but laminar retinal structure was still intact (**iii**). N = 16. (**b**) The presented sections illustrate the quality of retinae at the time point of isolation, at 6 and 14 d. On day one, all layers were distinguishable, and no signs of degeneration were seen (**iv**). On day 6, retinae look normal without significant signs of degeneration, though little cell loss might have occurred in the INL and ONL (black arrow) (**v**). POS were not visible. After 14 d of culture, laminar retinal structure was still intact (**vi**). N = 8. (**c**) Retinal thickness was determined in porcine and human retinae cultured for 14 d using micrographs of H&E-stained samples. It did not differ significantly over 14 d; an ANOVA excluded significant changes between time points for both porcine (pRetina) (*p* = 0.4882) and human (hRetina) tissue (*p* = 0.6126). Magnification: 400×; scale bar: 50 µm [44].

**Figure 9 antioxidants-12-01211-f009:**
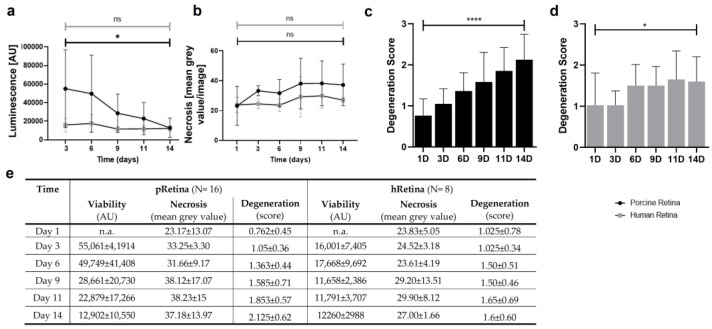
Cell viability, death, and degeneration in static culture of *p*- and hRetinae for 14 d. (**a**) The curves show luminescence measured in culture media of *p*- and hRetinae over 14 d. While luminescence remained low in hRetinae, it was high at the beginning in pRetinae, followed by a decrease. Luminescence inversely correlated to viability, indicating a high cell viability (or low cell death rate) for both species, except for within the first week in pRetinae. (**b**) Necrosis was determined by PI-staining, revealing low and stable rates of necrosis for both species from day 1 to 14. (**c**) Degeneration in cultured pRetinae increased over time but remained moderate with a maximum increase up to 2.13 (*p* < 0.0001). (**d**) In hRetina, degeneration remained stably low, with scores from 1.03 to 1.60 (*p* = 0.0290). (**e**) Individual values of viability, death, and degeneration analyses. * = *p* < 0.05; **** = *p* < 0.0001. [44].

**Figure 10 antioxidants-12-01211-f010:**
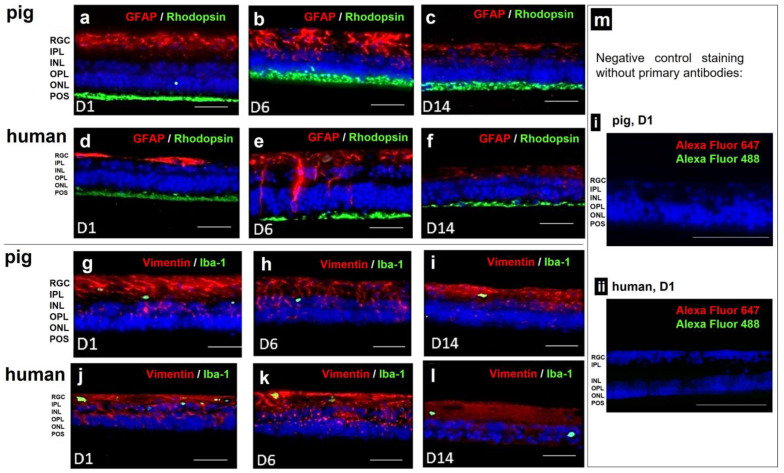
Immunohistochemical staining of p- and hRetinae from 1 to 14 d of culture. (**a**–**f**) Anti-GFAP-anti-rhodopsin double-staining confirmed POS preservation up to day 14 (green). GFAP staining revealed inflammatory reactions in Müller cells on day 6 of culture which decreased in the second week of culture below base staining (red). The pattern was similar for pRetinae and hRetinae. (**g**–**l**) Vimentin staining proved stable Müller cell presence and morphology for 14 d (red). Microglia activation, shown by Iba-1^+^ cell staining (green), remained low in both species at every time point. (**m**) The two micrographs (**i**,**ii**) show representative negative control staining without primary antibodies, but the two used secondary antibodies Alexa Fluor 647 (red) and Alexa Fluor 488 (green). N = 16 (pRetinae); N = 8 (hRetinae). Magnification: 200×; scale bar: 100 µm [44].

**Figure 11 antioxidants-12-01211-f011:**
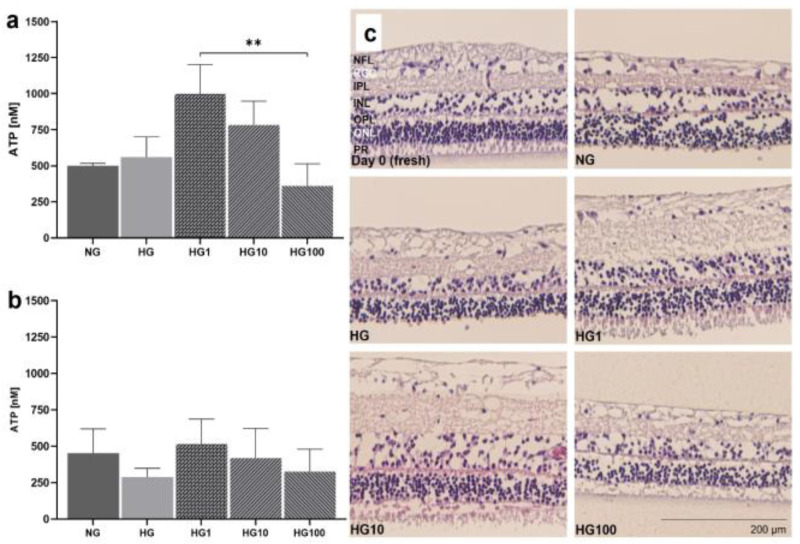
Anti-oxidative cell protection by scutellarin in HG conditions. pRetinae were cultured for 4 d in static conditions (Model 3) in NG (6 mM) or HG conditions (30 mM) and treated with scutellarin (1, 10 or 100 µM). (**a**) Viability determined on day 1 (D1) revealed comparable ATP level in both glucose conditions, but significantly increased viability after treatment with low (1 µM) or medium (10 µM) doses of scutellarin, while high doses (100 µM) had a detrimental effect. (**b**) On day 4 (D4), viability decreased in HG, but increased after addition of low and medium doses of scutellarin to levels of NG. (**c**) H&E staining of retinal sections fixed on day 4 of culture (plus one control fixed immediately after isolation: “fresh”), revealed a lowered degeneration in POS of scutellarin-treated samples if administered in low to medium doses. NG = normal glucose; HG = high glucose; HG1 = high glucose + 1 µM scutellarin; HG10 = high glucose + 10 µM scutellarin; HG100 = high glucose + 100 µM scutellarin. N = 4. Magnification: 200×; scale bar: 200 µm. ** = *p* < 0.01. [37].

**Figure 12 antioxidants-12-01211-f012:**
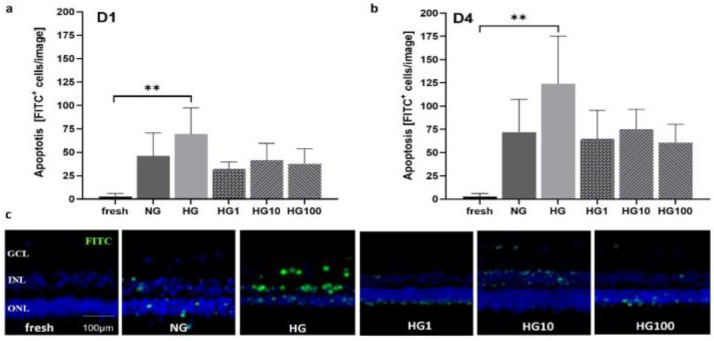
Apoptosis on day 1 and day 4 in HG and after treatment with the antioxidant scutellarin. Apoptosis was measured by a TUNEL assay in retinal cross sections cultured in NG (6 mM) or HG (30 mM) treated with 1, 10, or 100 µM scutellarin. (**a**) The number of apoptotic cells on day 1 increased in HG cultures compared to NG samples but decreased after scutellarin treatment, regardless of the dose. (**b**) On day 4, the rate of apoptosis was generally increased compared to day 1 with similar differences between groups. The highest number of apoptotic cells was counted in HG, followed by NG. In all scutellarin groups (HG1, HG10, HG100), apoptosis declined below NG values. (**c**) Micrographs show representative results on day 4 with increased numbers of apoptotic cells in HG, which could be decreased by scutellarin (HG1, HG10, HG100). It is obvious that in the HG and HG10 groups, most of the apoptotic cells were found in the INL, while in the NG, HG1, and HG100 groups, apoptosis was focused on the ONL. N = 4–6. Magnification: 200×; scale bar: 100 µm. NG = normal glucose; HG = high glucose; HG1 = high glucose + 1 µM scutellarin; HG10 = high glucose + 10 µM scutellarin; HG100 = 100 µM scutellarin. ** = *p* < 0.01. [37].

**Figure 13 antioxidants-12-01211-f013:**
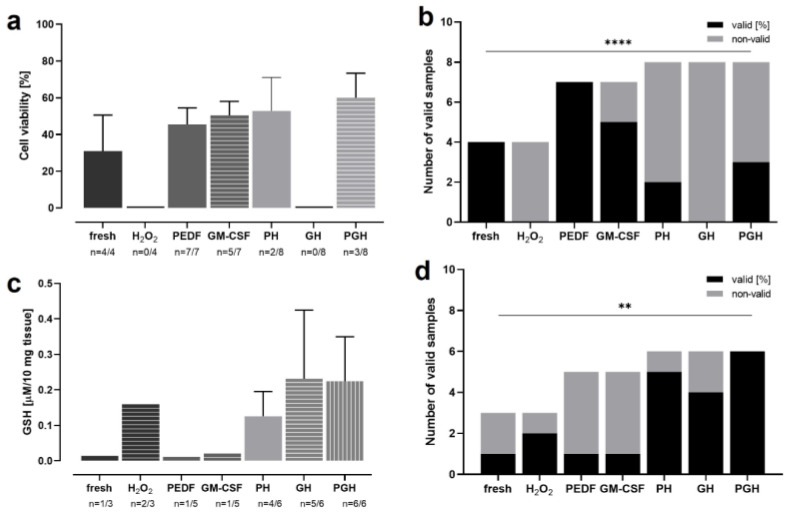
OS in rat retinae incubated with H_2_O_2_ and treated with PEDF and/or GM-CSF. Retinae were treated for 3 d with PEDF and/or GM-CSF before incubation with H_2_O_2_ on day 3. On day 4, viability and GSH levels were analyzed. Many samples were significantly degenerated and did not deliver technically valid results; thus, the number of valid results was analyzed as an additional parameter. (**a**) Highest viability was detected in tissue treated with both proteins (after H_2_O_2_ incubation), followed by the PH group, GM-CSF, and PEDF. Untreated, “normal” tissue without H_2_O_2_ incubation had second lowest level. In H_2_O_2_ and GH groups, no valid results were measured. (**b**) The number of valid viability measurements of data shown in (**a**) was increased in samples not H_2_O_2_ incubated and PEDF/GM-CSF treated. The number of valid results decreased in PH and PGH groups, and no valid results were received in H_2_O_2_ and GH groups. (**c**) Lowest GSH levels were measured in the fresh, PEDF, and GM-CSF groups. Medium levels were detected in H_2_O_2_ and PH groups, while highest GSH levels were determined in GH and PGH groups. (**d**) The percentages of valid GSH assay results of data shown in (**c**) were highest in groups PH, GH, and PGH, followed by H_2_O_2_. The lowest percentages were seen in groups fresh, PEDF, and GM-CSF. N = 3–8. PH = PEDF + H_2_O_2_; GH = GM-CSF + H_2_O_2_; PGH = PEDF + GM-CSF. A chi-square test for trend was performed, showing significant differences for the viability analysis (*p* < 0.0001) and for the GSH assay (*p* = 0.0069). ** = *p* < 0.01; **** = *p* < 0.0001.

**Table 1 antioxidants-12-01211-t001:** Comparison of presented retina culture models. The table summarizes the setup and the key features of Models 1 to 5.

Model		Species	Tissue	Type of culture	Processing	Medium	Temperature	Culture duration	Advantages	Disadvantages
1	bRetina	cattle	Choroid-RPE-retina	Static, customized grids	Scalpel	DMEM; Neurobasal, 2% B27	37 °C	4 d	Supporting tissue; easy available tissue	Complex isolation
2	pRetina, dynamic	pig	RPE-retina; retina	Dynamic, Minuth chamber	Scalpel	DMEM; Ames’	37 °C; 21 °C	3 d	Continuous medium flow; easy available tissue	More complex set up; only neural retina (+RPE)
3	pRetina, static	pig	Retina	Static, customized grids	Scalpel	Ames’	21 °C	4 d	Simple isolation; simple set-up; easy available tissue	Only neural retina
4	rRetina	rat	Retina	Static, inserts	Scalpel	Neurobasal, 2% B-27 & 1% N-2; Ames’; Ames’, 2% B-27 & 1% N-2	21 °C	9 d	Fresh tissue	Ethical concerns; only neural retina
5a	pRetina	pig	Retina	Static, inserts	Punch	Ames’ with 1% N2 & 1% B-27	21 °C	14 d	Low tissue damage; long culture; easy available	only neural retina
5b	hRetina	human	Retina	Static, inserts	Punch	Ames’, 1% N2 & 1% B-27	21 °C	14 d	Transferability; low tissue damage; long culture	Limited tissue availability; only neural retina

**Table 2 antioxidants-12-01211-t002:** Hemalum and eosin (H&E) staining protocol. The table lists the staining protocol, including the generally performed decreasing EtOH rehydration and increasing EtOH dehydration series.

Process	Reagents	Time	Temperature
Deparaffinization	Xylol	3 × 3 min	RT
Rehydration	EtOH 2× 100%, 96%, 90%, 80%, 70%, 50%, dd H_2_O	30 s each 3 min	RT
Staining	Hemalum	4 min	RT
	Tap H_2_O	3 × 2 min	
	Acid alcohol	10 s	
	dd H_2_O	2 × 1 min	
	Eosin	3 min	
	dd H_2_O	1 min	
Dehydration	50%, 70%, 80%, 90%, 96% 2 × 100% EtOH	30 s each 2 × 2 min	RT
Mounting	Mounting medium	1 min	RT

**Table 3 antioxidants-12-01211-t003:** Degeneration score. The score was created to quantify degeneration in BF micrographs from flat-mount preparations of retinal samples.

Grade	Tissue Quality
0	Healthy tissue, integral, no damage;
1	Irregular borders, small holes;
2	Big holes, patches of cell loss;
3	Big holes, major cell loss.

**Table 4 antioxidants-12-01211-t004:** Immunofluorescence staining protocol.

Process	Reagents	Time	Temperature
Deparaffinization	Xylol	3 × 3 min	RT
Rehydration	EtOH 100%, 96%, 90%, 80%, 70%, 50%, dd H_2_O	3 min each 3 × 5 min	RT
Demasking	Citrate buffer, 0.1 M pH 6 (boiling) cooling	15 min 30 min	95 °C–100 °C RT
Washing	PBS	3 × 10 min	RT
Blocking	PBS-BSA 3%	2 h	37 °C
Staining	Primary antibody (PBS-BSA 1%)	overnight	4 °C
Washing	PBS	3 × 10 min	RT
Staining	Secondary antibody (PBS)	30 min	37 °C
Washing	PBS	3 × 10 min	RT
Mounting	Mounting medium with DAPI	1 min	RT

**Table 5 antioxidants-12-01211-t005:** **Primary and secondary antibodies used for immunofluorescence staining.** GFAP: Glial fibrillary acidic protein; Iba-1: Ionized calcium-binding adapter molecule 1.

Specificity	Cells Labelled	Host	Conjugate	Clonality	Company	ID	Dilution
**Primary antibodies**							
Protein kinase C	Bipolar cells	Chicken	Unconjugated	Polyclonal	Abcam * ab14078	n.a.	1:50
Rhodopsin	Rods	Rabbit	Unconjugated	Polyclonal	Novus Biological ^$^ NLS1052	AB_2178795	1:700
GFAP	Astrocytes	Mouse	Unconjugated	Monoclonal	Millipore ^†^ MAB360	AB_11212597	1:300
Iba-1	Microglia	Rabbit	Unconjugated	Polyclonal	Fujifilm Wako Pure Chemical Corporation ^§^ 01-1874	AB_2314666	1:750
Vimentin	Müller cells	Mouse	Unconjugated	Monoclonal	Merck^†^ MAB3400	AB_94843	1:120
**Secondary antibodies**							
Mouse	NA	Donkey	Alexa Fluor 647	NA	Abcam ab150107	AB_2890037	1:250
Rabbit	NA	Donkey	Alexa Fluor 488	NA	Jackson ^#^ 711546152	AB_2340619	1:100

* Cambridge, UK; ^$^ Zug, Switzerland; ^†^ Darmstadt, Germany; ^§^ Osaka, Japan; ^#^ Ely, UK.

## Data Availability

The data presented in this study are openly available on the Zenodo repository doi: 10.5281/zenodo.7974096, uploaded on 26 May 2023.

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
