# Peer review of "Mammalian Animal and Human Retinal Organ Culture as Pre-Clinical Model to Evaluate Oxidative Stress and Antioxidant Intraocular Therapeutics"

_antioxidants, 2023, doi:10.3390/antiox12061211_

Round 1

Reviewer 1 Report

The authors of the manuscript ID:2416218, title: Mammalian animal & human retinal organ culture as pre-clinical model to evaluate oxidative stress and antioxidant intraocular therapeutics.
In manuscript submitted for review, different retina organ culture models were evaluated, and their advantages and drawbacks as preclinical test system discussed.
 The manuscript is well designed experimentally as well as substantively prepared. The introduction broadly discusses two diseases, age-related macular degeneration (AMD) and diabetic retinopathy (DR), which impair vision and can lead to blindness. Isolation and cultivation of various retina models was carried out. Results: Twelve figures containing the results as well as color graphics from the Immunohistochemical staining of animals and human retinae research well illustrate the course and results of the research.

It is not clear why the authors placed table 5 as well as the description of the methods between the discussion and the conclusions. Please correct, add to the methods section.

Discussion and references: The authors discussed the results using professional literature. Including the introduction, 66 items of international literature were used.

Author Response

The authors of the manuscript ID:2416218, title: Mammalian animal & human retinal organ culture as pre-clinical model to evaluate oxidative stress and antioxidant intraocular therapeutics.
In manuscript submitted for review, different retina organ culture models were evaluated, and their advantages and drawbacks as preclinical test system discussed.
  The manuscript is well designed experimentally as well as substantively prepared. The introduction broadly discusses two diseases, age-related macular degeneration (AMD) and diabetic retinopathy (DR), which impair vision and can lead to blindness. Isolation and cultivation of various retina models was carried out. Results: Twelve figures containing the results as well as color graphics from the Immunohistochemical staining of animals and human retinae research well illustrate the course and results of the research.

We thank the reviewer for his careful review and kind evaluation!

It is not clear why the authors placed table 5 as well as the description of the methods between the discussion and the conclusions. Please correct, add to the methods section.

The purpose was to show table 5 as a summary of the presented models and data in the discussion to facilitate final result discussion and conclusion. However, we understand very well the comment of the reviewer 1 and agree that a presentation in the methods and material section might be more appropriate. Thus, table 5 (now table 1) has been shifted to section 2.2.6, this and following table numbers have been adapted, and the section numbers have been updated. According to reviewer 2, we separated table 1 and associated photographs, which are now presented separately as figure 1.

(lines 269-286, revised manuscript in track mode)

Discussion and references: The authors discussed the results using professional literature. Including the introduction, 66 items of international literature were used.

We thank the reviewer for his appreciation that we used professional, international and sufficient literature to present and discuss our study.

We would like to add that we used the review process to read our manuscript again carefully and corrected a few typing errors. Also, we added the now available DOI on which the raw data will be available (to be released on June 30, 2023).

Please find the revised manuscript in the attachment.

Reviewer 2 Report

The manuscript needs some more extra work to be reconsidered for a publication as good quality. The following recommendations aims to improve the readability and the quality of this manuscript:

1-               FIGURE 1C: to me is not visible the GFAP signal in the astrocytes at 0d that should be visible. Furthermore, the figure legend miss the abbreviation for RPER, RGC and INL, please add.

2-               FIGURE 2: please explain the meaning of the black and the red arrows. Please add also in the figure legend the arrows’ information.

3-               Regarding the cell count’s plot please always use the same scale for the y axis

4-               FIGURE 4C: Please add the abbreviation for the ILM and ELM in the figure legend.

5-               FIGURE 5F: it is difficult to see the legend with the chosen colours please use a more readable legends in term of colour\ grey scale.

6-               FIGURE 5B, C, D and E: Where are the white, red and black arrows? I cannot see in the figure, please add.

7-               FIGURE 9: here is my main concern in term of IHC quality, I would recommend to the authors to add a more clear, visible images; to me the red signal is often a background (i.e., a-D1, g-D1 and i-D4). Please also add the negative control images as supplementary information to exclude any background images acquisition. The IBA1 signal is not appreciable, and I cannot see the proper microglia structure in all images presented, please use different with high quality images. Please correct the layers’ name, they are not alienated with the blue staining.

8-               FIGURE 11: in my opinion would be better to show and TUNEL+ cell counting related to the different retinal layers and the results should be explained and discussed in the proper section.

9-               TABLE 5: please separate from the below figure (that will be FIGURE 12) and add the corresponding figure legend; please increase the size of the presented images.

Author Response

The manuscript needs some more extra work to be reconsidered for a publication as good quality. The following recommendations aims to improve the readability and the quality of this manuscript:

We thank the reviewer for his careful review and hope that our modifications can clarify all open questions.

Please note that the figures had to be renumbered. Thus, the numbers of the figures in our answers (corresponding to the new version to the manuscript), do not correspond to the numbers of the figures in the reviewers’ comments. We hope that the retrieval of the modified text and figures will nevertheless be feasible.

The line numbers refer to the updated manuscript in the track mode (please see in the attachment).

  1. FIGURE 1C: to me is not visible the GFAP signal in the astrocytes at 0d that should be visible. Furthermore, the figure legend miss the abbreviation for RPER, RGC and INL, please add.

We agree to the comment and added the explanations for the missing abbreviations. Regarding the GFAP signal at 0d (which is indeed very low), we recognized that the associated text in figure caption 2 (and the main text) was confusing and modified it. We hope that the new text is better understandable:

“GFAP expression remained low in retinae at 4 d in culture (bottom, 4 d), comparable to samples stained directly after isolation (top, 0 d). White arrows indicate areas of weak GFAP expression.”

Additionally, we increased the size of the figure to improve visibility of low GFAP signal at 0d.

(lines 447-459)

2. FIGURE 2: please explain the meaning of the black and the red arrows. Please add also in the figure legend the arrows’ information.

We apologize for the missing information. The black and red arrows indicate areas of retinal degeneration in different retinal layers. The information has been added to the figure caption: “Overall, the “+RPE” culture decreased in thickness and revealed holes and cell loss as indicated by the black arrow in the INL and the red arrow in the POS layer.” (lines 481-482)

3. Regarding the cell count’s plot please always use the same scale for the y axis

We are sorry for this incoherence and uniformed the scale in figure 5. Now, all cell counts figures (nos. 3, 4 and 5) uniformly show the same scale for the y axis from 0-300 cell number [cells/image] with an increment of 100. (line 511)

4. FIGURE 4C: Please add the abbreviation for the ILM and ELM in the figure legend.

We added the missing abbreviations in line 520 of the revised manuscript.  

5. FIGURE 5F: it is difficult to see the legend with the chosen colours please use a more readable legends in term of colour\ grey scale.

We agree that the chosen grey scales in figure 6f might be difficult to differentiate and modified them accordingly. (line 530)

6. FIGURE 5B, C, D and E: Where are the white, red and black arrows? I cannot see in the figure, please add.

We are very sorry having overseen this mistake. It seems that by saving the composed figure as tif-image, the arrows got lost. This has been corrected and the entire figure can now be seen in line 530.

7. FIGURE 9: here is my main concern in term of IHC quality, I would recommend to the authors to add a more clear, visible images; to me the red signal is often a background (i.e., a-D1, g-D1 and i-D4). Please also add the negative control images as supplementary information to exclude any background images acquisition. The IBA1 signal is not appreciable, and I cannot see the proper microglia structure in all images presented, please use different with high quality images. Please correct the layers’ name, they are not alienated with the blue staining.

Thank you for this valuable comment. We hopefully answered satisfyingly the critics related to the figure 10:

  • Figure quality has been improved and the figure size increased.
  • We added a panel of negative controls, one per species and double staining, to exclude background staining acquisition.
  • We hope that the improved quality and increased size now better allows to see the less ramified, amoeboid shape of activated Iba-1+ microglia.
  • Layer labelling has been better aligned to the blue DAPI staining.

(line 623 and lines 629-635)

8. FIGURE 11: in my opinion would be better to show and TUNEL+ cell counting related to the different retinal layers and the results should be explained and discussed in the proper section.

We agree to the reviewer’s comment that the number of apoptotic cells in the different groups showed a different distribution pattern in the retinal layers. We added a respective paragraph that relates the results to the different retinal layers in lines 673-679 and modified the caption of figure 12 accordingly (lines 680-685 and 695-697).

9. TABLE 5: please separate from the below figure (that will be FIGURE 12) and add the corresponding figure legend; please increase the size of the presented images.

We thank the reviewer for this comment. According to reviewer 1, table 5 (now table 1) has been shifted to section 2.2.6, this and following table numbers have been adapted, and the section numbers have been updated. Additionally, to answer the current request, we separated table 1 and associated photographs, which are now presented enlarged and separately as figure 1 (lines 269-286).

We would like to add that we used the review process to read our manuscript again carefully and corrected a few typing errors. Also, we added the now available DOI on which the raw data will be available (to be released on June 30, 2023).

Round 2

Reviewer 2 Report

I would like to thanks the authors for the extra work done for this manuscript. Overall, I think that the paper is now suitable for publication.